# Development of an HRMA-Based Marker Assisted Selection (MAS) Approach for Cost-Effective Genotyping of *S* and *M* Loci Controlling Self-Compatibility in Apricot (*Prunus armeniaca* L.)

**DOI:** 10.3390/genes13030548

**Published:** 2022-03-20

**Authors:** Bianca Maria Orlando Marchesano, Remo Chiozzotto, Irina Baccichet, Daniele Bassi, Marco Cirilli

**Affiliations:** Department of Agricultural and Environmental Sciences (DISAA), University of Milan, 20133 Milan, Italy; bianca.orlandomarchesano@studenti.unimi.it (B.M.O.M.); remo.chiozzotto@unimi.it (R.C.); irina.baccichet@unimi.it (I.B.); daniele.bassi@unimi.it (D.B.)

**Keywords:** MAS, apricot, HRMA, self-compatibility, *S*-locus, *M*-locus

## Abstract

The apricot species is characterized by a gametophytic self-incompatibility (GSI) system. While GSI is one of the most efficient mechanisms to prevent self-fertilization and increase genetic variability, it represents a limiting factor for fruit production in the orchards. Compatibility among apricot cultivars was usually assessed by either field pollination experiments or by histochemical evaluation of in vitro pollen tube growth. In apricots, self-compatibility is controlled by two unlinked loci, *S* and *M*, and associated to transposable element insertion within the coding sequence of *SFB* and *ParM-7* genes, respectively. Self-compatibility has become a primary breeding goal in apricot breeding programmes, stimulating the development of a rapid and cost-effective marker assisted selection (MAS) approach to accelerate screening of self-compatible genotypes. In this work, we demonstrated the feasibility of a novel High Resolution Melting Analysis (HRMA) approach for the massive screening of self-compatible and self-incompatible genotypes for both *S* and *M* loci. The different genotypes were unambiguously recognized by HRMA, showing clearly distinguishable melting profiles. The assay was developed and tested in a panel of accessions and breeding selections with known self-compatibility reaction, demonstrating the potential usefulness of this approach to optimize and accelerate apricot breeding programmes.

## 1. Introduction

Hermaphrodite plants developed physical and chemical strategies to prevent self-fertilization and to promote outcrossing for increasing genetic variability [1]. Self-incompatibility (SI) is considered one of the most efficient mechanisms to induce outcrossing. Gametophytic Self-Incompatibility (GSI) is the most diffused SI strategy in angiosperms. The GSI mechanism acts through the inhibition of self-pollen tube growth in the style [2]. The self- or non self-pollen recognition is driven by the haploid pollen genome (i.e., if one of the two pistil *S* alleles matches with the pollen one, the pollen is recognized as self and the pollen tube growth is arrested) [3,4]. GSI specific recognition is controlled by a multi-allelic locus (*S*-locus), containing at least two linked genes: an *S-RNase* and a *S*-locus F-box (*SFB*), respectively, expressed in pollen and pistil [5].

Apricot (*P. armeniaca* L.) is a diploid species belonging to the genus *Prunus* in the Rosaceae family. Although traditional European apricot cultivars are self-compatible, the implementation of North American materials, used for introducing PPV resistance have caused an increasing spread of incompatible varieties [6]. The knowledge of pollination requirements of existing or newly introduced apricot cultivars is crucial for growers, with cross-pollinations being a major determinant of fruit set [7]. The planting of SI cultivars requires pollen donors (‘pollinizers’), usually interspersed throughout orchards (or multi-cultivars systems). Indeed, cultivars can be classified as incompatible (when parents share both *S*-haplotypes), semi-compatible (one shared *S*-haplotype) or fully compatible (no shared *S*-haplotype). Moreover, semi-compatibility may have a significant impact on fruit set reduction, since half the available pollen grains are rejected. Incompatibility relationships between cultivars were traditionally assessed by controlled pollination in the field (although pollen viability and pollination could be strongly influenced by the environment) or by histochemical evaluation of in vitro pollen tube growth [8,9]. PCR-based tools have been developed to determine the molecular determinants of SI, allowing the genotyping of allelic composition at *S*-locus [10,11,12,13]. Currently, at least 17 incompatibility groups and 33 *S*-alleles have been identified for *S-RNase* gene (including the SC allele) [14,15].

At the genetic level, SC in apricot germplasm is controlled by two unlinked loci, *S* and *M* [16]. Locus *S* self-compatibility is conferred by a 358 bp insertion within the *SFB* coding sequence [17]. The *SFB* gene is composed of a single F-box motif, two variable and two hypervariable regions. The F-box is located at the N-terminal region, and highly conserved among different *Prunus* species; on the contrary, one variable and two hypervariable regions are located at the C-terminus and are considered as responsible for the SFB specificity [11]. The insertion sequence holds a high percentage of identity with a transposable element and induces a premature stop codon in the ORF of SC-associated *SFB* allele and the breakdown of the GSI activity. The *M*-locus represents another source of SC found in cultivars such as ‘Canino,’ ‘Katy’, and ‘Portici’ [17,18]. This locus has been fine-mapped within a ~134 kb region at the distal end of chromosome 3, identifying a 358 bp MITE insertion causing a premature stop codon within the candidate gene *ParMDO*, a putative disulfide bond A-like oxidoreductase [16,19]. For both loci, some SC-allele specific PCR markers were provided, although their performance was not tested in a wider genetic background, nor optimized for a high-throughput marker assisted selection (MAS) application. High Resolution Melting Analysis (HRMA) is a well-established approach for the identification of polymorphisms within PCR amplicons based on their melting temperature (T_m_) profile [20,21]. The benefits of this technique in terms of both time, cost efficiency and assay sensitivity and sensibility are largely described in various fruit tree species [20,21,22,23].

In this work, a rapid and cost-effective MAS strategy based on High Resolution Melting Analysis (HRMA) was developed to genotype target SC alleles at *S* and *M* loci in the apricot. The performances and usefulness of these assays were validated in a panel of accessions and selections from various progenies in the framework of the MAS.PES breeding program (as shown in Table 1, and in Appendix A).

## 2. Materials and Methods

### 2.1. Plant Material, Phenotyping and DNA Extraction

Young leaves of 203 accessions including cultivars and breeding selections from the MAS.PES project (https://sites.unimi.it/maspes/ (accessed on 3 January 2022)) were sampled from ASTRA-Mario Neri experimental farm (Imola, Bologna, Italy). Among all the plants, 56 accessions (Table 1) were used for the development and validation of PCR and HRM assays for both *S* and *M* loci, while the remaining plants were used for assay application (Appendix A). Accessions with unknown SC reaction were tested by self-pollination in the field. Briefly, several branches (at least 200–300 flower buds) or the whole tree were covered with insect-proof bags or cages to prevent cross pollination, recording fruit set percentage. DNA was extracted following a previously described protocol [20] optimized from the original Doyle and Doyle protocol (1987) [34]. Briefly, 2 mg of leaf tissue were ground using a TissueLyser (Qiagen). Then, 900 µL of 65 °C preheated CTAB isolation buffer (with 0.2%—mercaptoethanol) was added to the lysate tissue and incubated for 20 min at 65 °C shaking gently 3 times. After, 800 µL chlorophorm:isoamyl alcohol (24:1) was added and mixed by inverting the tube. Tubes were centrifugated for 5 min at 10,00 rpm at room temperature; 700 µL of the supernatant was transferred in a clean 1.5 mL tube and 750 µL of room temperature IPA were added. Tubes were centrifugated for 20 min at 14,000 rpm at 10 °C. The supernatant was discarded and 800 µL of 70% ethanol was added. The pellet was detached from the tube bottom. After centrifugation for 10 min at 14,000 rpm at 10 °C, the supernatant was discarded, and the ethanol was evaporated. Finally, 110 µL of water was added. DNA extraction was checked by electrophoresis of 10 µL of the sample on agarose gel.

### 2.2. Primer Design

In this work, both previously developed primers [11,13,17] and newly designed ones were tested to identify the most suitable for HRM assay development. Specific primers were designed by using Primer3 software (https://primer3.ut.ee/ (accessed on 3 February 2021)) (Table 2. For *S* locus design, publicly available sequences of *SFB* and *ParMDO* were retrieved from NCBI and aligned on Multalin software (http://multalin.toulouse.inra.fr/multalin/ (accessed on 3 February 2021)) in order to find the most conserved regions among the various alleles. Specific primers complementary to the left and the right insertion borders (*SFB-F//SFB OUT-R*) were developed to allow the amplification of both the *SFB* and the *SFBc* alleles. To univocally detect the presence of the insertion within the *SFB* gene, primers *SFB INS-F* and *SFB INS-R* were designed within the insertion sequence. For *M* locus, specific primers for the amplification of the different alleles were developed starting from the sequences deposited at the GenBank (accession numbers KY429940 and KY429941). *LocusM-F* and *LocusM-R* primers were designed to anneal to the left and right insertion borders, respectively.

### 2.3. PCR Amplification and HRMA Analysis

PCR was performed in a 96-well plate containing 12.5 µL GoTaq^®^ (Promega), 1 µL of each primer, 20 ng of genomic DNA and distilled water to reach a final volume of 20 µL. The amplification was carried out starting from an initial denaturing temperature of 95 °C for 3 min followed by 32 cycles at 94 °C for 30 s, 60 °C for 40 s and 72 °C for 50 s; and a final extension at 72 °C for 5 min. PCR products were visualized using 1.5% agarose gel electrophoresis, evaluating the amplicons size using a 100 bp ladder (PCRBIO Ladder IV). HRM was performed in a 96-well plate in a final volume of 20 µL containing 10 µL HRM Mix (2x qPCRBIO, PCRBIOSYSTEM), 0.7 µL of each primer, 20 ng of genomic DNA and distilled water to a final volume of 20 µL. HRMs were carried out using a QuantStudio 3 (Applied Biosystem) with the following cycling conditions: 5 min at 95 °C, 40 cycles for 10 s at 95 °C, 20 s at primer specific annealing temperature, 30 s at 72 °C, followed by an HRM stage of 15 s at 95 °C, 1 min at 60 °C and a final melting step at a 60–95 °C gradient with 0.025 °C/s ramp rate. Melting data were normalized according to the operator manual analysis of both the amplification plot and the derivative melting curves. HRM products were run on 1.5% agarose electrophoresis gel to verify amplicon size.

## 3. Results

### 3.1. HRMA Assay for SFB_C_ Allele at S-Locus

As previously reported, the *S*-locus self-compatible allele carries an insertion of a transposable element at position +904 to 1261 bp from the ATG start codon, leading to a truncated protein lacking 75 amino acids and the two hypervariable regions [17]. With the purpose of developing a novel HRMA-based molecular assay able to distinguish heterozygous and homozygous *SFB_C_* genotypes (i.e., able to amplify either the *SFB_C_* and the other various *SFB* alleles), available nucleotide sequences of *SFB/SFB_C_* alleles were retrieved and aligned to find the most conserved regions for primers design, and amplify as many alleles as possible. However, the alignment showed the presence of a high number of polymorphisms, particularly in the region downstream the 358 bp insertion (Appendix A). Moreover, the size of transposon insertion further narrows the candidate regions compatible with an HRMA assay (i.e., less than 500–600 bp). Therefore, several primer pairs were developed (Table 2) and tested in a subset of 56 accessions with known *S*-locus genotype (*S-RNAse* and/or *SFB*) and SI/SC phenotype, including homozygous or heterozygous SC for *S_C_/SFB_C_* allele and SI accessions with different *S*-allele assortments (Table 1). Among the several pairs of primers, *SFB-F* and *SFB-OUT-R* resulted as the most suitable combination, able to amplify both the compatible (*SFB_C_*) and incompatible (*SFB_I_*) allele(s) (amplicon size of ~600 and ~200 bp, respectively) with a clear distinction between homozygous (*SFB_C_*_-HOM_) and heterozygous (*SFB_C-_*_HET_) SC genotypes (Figure 1). Similarly, SC and SI genotypes were unambiguously recognized by the HRMA assay through a different melting curves profile. Indeed, derivative melting curves analyses showed a single peak for *SFB_C-_*_HOM_, with an average T_m_ of 78.5 ± 0.2 °C, and two peaks with broadened melting shape for *SFB_C_*_-HET_ with average T_m_ of 78.45 ± 0.1 °C and 79.56 ± 0.1 °C, respectively. As expected from the high genetic polymorphisms across *SFB_I_* alleles, different melting profiles were detected in self-incompatible genotypes, although still clearly distinguishable from either *SFB_C_*_-HOM_ or *SFB_C_*_-HET_ (Figure 1). Electrophoresis gel of HRMA amplicons showed the absence of *SFB_C_* band in SI genotypes, while confirming its presence in either homozygous or heterozygous ones (Figure 1 and Appendix A). In addition to the *SFB-F* and *SFB-OUT-R*, another assay was developed (*SFB-INS-F* and *SFB-OUT-R*) with the purpose of genotyping only the *SFB_C_* allele, irrespective of the distinction between homozygous and heterozygous genotypes. This kind of assay may turn out to be useful for MAS, in case the cross-parent genotypes are a priori known. For this assay, HRMA showed a clear peak with a T_m_ of 77.8 ± 0.05 °C in SC genotypes, in contrast to the noised or absent melting profile in SI ones (Figure 2). Even for this assay, electrophoresis gel of HRMA amplicons showed a fragment of the expected size of ~300 bp only in SC genotypes (Figure 2).

### 3.2. HRMA Assay for MDO-m Allele at M-Locus

The *M*-locus self-compatible allele carries an insertion of a transposable element at position +332 to +690 bp from the ATG start codon [19]. A new pair of primers suitable for HRMA-based assay was developed (*LocusM-F* and *LocusM-R*) (Figure 3A) which allowed a clear distinction between the compatible (*m*) and incompatible (*M*) alleles. Since any homozygous (*m*_-HOM_) genotype was found in our germplasm collection, an *m*_-HOM_ control was generated through the gel excision and purification of *m* allele. As shown in Figure 3, homozygous (*m*_-HOM_) control and heterozygous (m_-HET_) genotypes were unambiguously recognized by HRMA, showing clearly distinguishable melting profiles. The *m* allele showed a single peak with a T_m_ of 77.9 ± 0.5 °C (red colored curve) while the heterozygous *m* (green) and *M* (blue and yellow) allelic assortments showed clearly distinguishable broadened melting shapes (Figure 3). Electrophoresis gel of HRMA amplicons confirmed the presence of *m* alleles in all SC genotypes and, conversely, their absence in SI ones (Table 1 and Appendix A).

## 4. Discussion and Conclusions

The characterization of genetic bases controlling self-compatibility has opened the possibility for the development of molecular-based approaches to exploit this crucial trait for apricot breeding. At *S*-locus, assessment of either self-compatibility and/or incompatibility groups still relies on the genotyping of *S-RNase* alleles by amplicons size-based identification, a laborious technique involving capillary electrophoresis analysis. Although *S-Rnase* genotyping is still useful for characterizing allelic composition in SI cultivars, this molecular tool is not adequately efficient at specifically selecting for SC trait. Moreover, the use of SC-linked *S-RNAse* amplicon (*S_C_*) to discriminate SC from SI cultivars can lead to incongruent attribution, as for example the identical coding regions of the *S_8_*- and *S_C_-RNAse* alleles [15], or in the case of a linkage-breaking event. An *SFB_C_* allele-specific PCR assay suitable for germplasm screening was previously developed by Vilanova et al. (2006) [17] (*RFBc-F//SFBins-R*) based on the consensus sequence of the *Prunus SFB* alleles [11]. Unfortunately, the amplicon size (over 1000 bp) was incompatible with a real-time PCR analysis. Moreover, *RFBc-F//SFBins-R* primers were unable to amplify the SFB allele in all cultivars, probably due to the high number of polymorphisms within the SFB sequences.

The *M*-locus represents an *S*-locus independent source of self-compatibility identified in cultivars ‘Canino’ and ‘Katy’ [17,18]. Similar to the *SFBc* allele, a MITE insertion induces a premature stop codon in the compatible *ParMDO* allele, putatively leading to a truncated protein that lacks four of the six exons [19]. Primers for the *m* (self-compatible) allele-specific genotyping by PCR assay have already been developed by Muñoz-Sanz (2017) [16] (*RT594-F//RT594-R*). However, even for this marker, the amplicon size (over 1000 bp) was not suitable for an HRMA-based assay.

The newly developed HRMA assays were able to unequivocally discriminate SC from SI genotypes, as revealed by clear distinctions between the melting profiles. A wide apricot germplasm was used for assay testing and validation, with the aim of including a wide genetic variability. Results were largely in agreement with genotypes/phenotypes reported in the literature, although some inconsistencies were observed (Table 1): in the case of ‘Harleyne’, characterized by the *S_3_/S_20_ S-RNAase* allele, the presence of the heterozygous *SFB_C_* allele is congruent with its well-known SC phenotype [35], suggesting the hypothesis of a linkage breaking between *SFB_C_* and *S_C_* in the *S*-locus haplotype of this accession; in ‘Dorada’, ‘Ladycot’ and ‘Mirlo Naranja’, a heterozygous *SFB_C_* allele was detected in contrast to the homozygous *S_C_/S_C_* genotype reported in the literature, probably due to problems with the amplification of the incompatible(s) *S-RNAase* allele(s) [14]; in ‘Pricia’, an SC cultivar heterozygous for the *S_C_* allele, the *SFB_C_* allele was not detected although these cultivars were heterozygous for *m*_-HET_ genotype; in this case, an erroneous attribution of the cultivars or a sampling error could provide a possible explanation.

In conclusion, the obtained results clearly demonstrate the feasibility of an HRMA-based approach for rapid and cost-effective genotyping of self-compatibility at loci *S* and *M*. The possibility of unequivocally discriminate SC from SI genotypes represents a highly valuable tool for optimizing marker-assisted selection of self-compatibility trait and for accelerating apricot breeding programmes.

## Figures and Tables

**Figure 1 genes-13-00548-f001:**
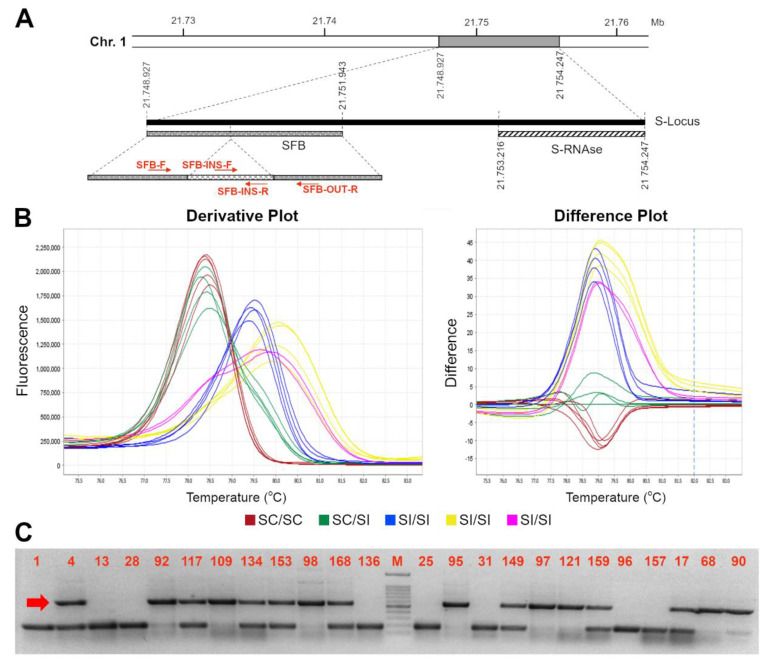
High resolution melting assay for distinction of SFBc genotypes. Schematic representation of the *S*-locus (**A**). On the left, the *SFB* gene with the primer used in this work is represented; the light grey line represents the 358 bp inserted element at position 904 to 1.261 from the first nucleotide of the ORF [17]; (**B**) Derivative (left) and difference (right) plots of melting curves from HRMA assays for *SFB* alleles by using the *SFB-F//SFB OUT-R* couple of primers: in blue the SC *SFB*-_HOM_, in green the *SC*-_HET_ and in red the SI profiles; (E,F) different possible *SFB* SI homozygous curves; (**C**) Electrophoresis gel of HRMA product showing *SFBc* (indicated by the red arrow) and *SFB_i_* allele amplicons of ~600 and ~200 bp long, respectively. A 100 bp marker has been used as ladder in the analysis. Accession number is indicated in Table 1.

**Figure 2 genes-13-00548-f002:**
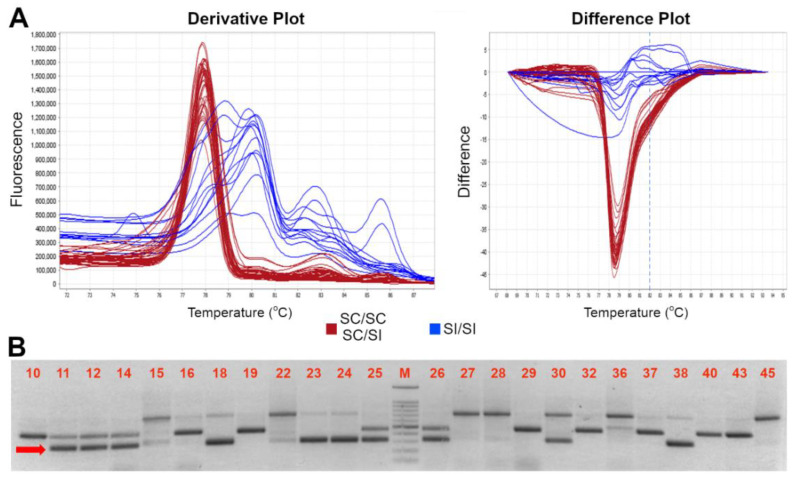
High resolution melting assay for distinction between SC and SI genotypes. (**A**) Derivative (left) and difference (right) plot profiles of the HRMA assay of the *S*-locus *SFB* specific amplification by using the *SFB INS-F//SFB OUT-R* primers; the red and blue color indicated the two different profiles of SC and SI profiles, respectively. (**B**) Electrophoresis gel of HRMA product showing the *S_C_* allele (indicated by the red arrow) of ~300 bp long. A 100 bp marker has been used as ladder in the analysis. Accession number is indicated in Table 1.

**Figure 3 genes-13-00548-f003:**
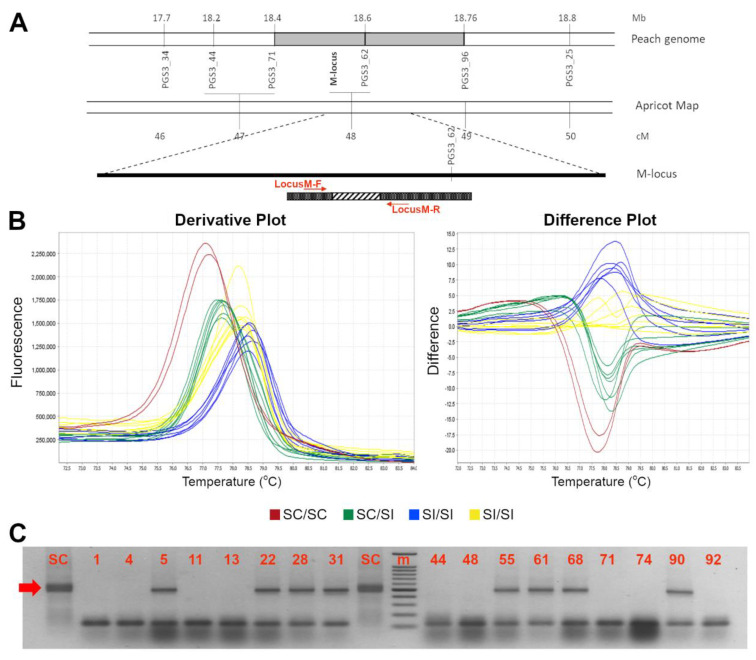
High resolution melting assay for distinction between SI (M), and SC (*m-*_HOM_ and *m*-_HET_) genotypes. Schematic representation of the *M*-locus (**A**). At the bottom, the *ParM-7* (*ParMDO* gene) with the primer used in this work; the diagonal stripe pattern represents the 358 bp inserted element at position 332 to 690 from the first nucleotide of the ORF. (**B**) Derivative (left) and aligned (right) melting curves of the HRM assays for the *M*-locus alleles: in red and yellow two possible SI homozygous profiles, in blue the SC *m*-_HOM_ and in green the *m*-_HET_. (**C**) Electrophoresis gel of HRMA product showing SC (indicated by the red arrow) and SI alleles amplicons of ~500 and ~150 bp long, respectively. A 100 bp marker has been used as ladder in the analysis. Accession number is indicated in Table 1.

**Table 1 genes-13-00548-t001:** List of accessions used for locus *S* and *M* HRMA assays development and validation. Genotype (also including *S-RNAse* alleles as reported in literature), phenotype and related references are indicated. Locus *S* and *M* alleles obtained in this work are listed in Locus *S SFB* and Locus *M ParMDO* columns respectively. *Asterisks marks accessions in disagreement with previous literature data.

#	Accession	Floral Compatibility	Locus *S* SRNAse	Locus *S* SFB	Locus *M* ParMDO	References
1	Yamagata	SI	*S8/S-*	*SI/SI*	*M/M*	[24]
4	Dorada *	SC	*Sc/Sc*	*SC/SI*	*M/M*	[1]
5	Dama Rosa	SC		*SC/SI*	*m/M*	[25]
11	Moixent	SC		*Sc/SI*	*M/M*	[26]
13	Estrella	SI		*SI/SI*	*M/M*	[27]
17	Mediabell	SC	*S6/Sc*	*SC/SI*	*M/M*	[14]
19	San Castrese	SC	*Sc/S-*	*SC/SI*	*M/M*	[24]
20	Gilgat	SI		*SI/SI*	*M/M*	
22	Dama Taronja	SC		*SI/SI*	*m/M*	[25]
24	Toni	SC		*SC/SI*	*M/M*	[27]
25	Congat	SI		*SI/SI*	*M/M*	
28	Pricia *	SC	*Sc/S3*	*SI/SI*	*m/M*	[14]
31	Pelese di Giovaniello	SC	*S1/S2*	*SI/SI*	*m/M*	[28]
38	Mirlo Naranja *	SC	*Sc/Sc*	*SC/SI*	*M/M*	[14]
42	Harostar	SI		*SI/SI*	*M/M*	[29]
43	Bella di Imola	SC		*SC/SC*	*M/M*	[30]
44	Harval	SC		*SC/SI*	*M/M*	[31]
48	Pieve	SC	*Sc/S1*	*SC/SI*	*M/M*	[24]
55	Murciana *	SC	*Sc/Sc*	*SC/SI*	*m/M*	[1,14]
59	Frisson	SC		*SC/SC*	*M/M*	[32]
61	Dama Vermella	SC		*SC/SI*	*m/M*	[25]
68	Farfia	SC	*Sc/Sc*	*SC/SC*	*m/M*	[14]
69	Spring Blush	SI	*S3/S8*	*SI/SI*	*M/M*	[14]
71	Pisana	SC	*S2/Sc*	*SC/SI*	*M/M*	[28]
74	Fiamma	SC	*Sc/Sc*	*SC/SC*	*M/M*	
78	Aurora	SI	*S1/S17*	*SI/SI*	*M/M*	[16]
79	Faralia	SC	*Sc/S6*	*SC/SI*	*M/M*	[14]
85	SEO	SI	*S6/S9*	*SI/SI*	*M/M*	[16]
86	Goldrich	SI	*S1/S2*	*SI/SI*	*M/M*	[10]
90	Farbaly	SC	*Sc/Sc*	*SC/SC*	*m/M*	[14]
92	Ninfa	SC	*Sc/S7*	*SC/SI*	*M/M*	[16]
93	Amabile Vecchioni	SC		*SC/SI*	*M/M*	[31]
95	Tondina di Tossignano	SC		*SC/SC*	*M/M*	
96	Cegledi	SI	*S8/S9*	*SI/SI*	*M/M*	[16]
97	Sulmona	SC		*SC/SC*	*M/M*	[30]
98	Trzii Bucresti	SC		*SC/SC*	*M/M*	
106	Mono	SC		*SI/SI*	*m/M*	[31]
109	Tyrinthos	SC	*Sc/Sc*	*SC/SC*	*M/M*	[16]
114	Magyar Kaiszi	SC		SC/SI	*M/M*	[30]
117	Lito	SC	*Sc/S6*	*SC/SI*	*M/M*	[16]
118	Royal Roussillon	SC		*SC/SI*	*M/M*	[32]
119	Harcot	SI	*S1/S4*	*SI/SI*	*M/M*	[18]
120	Reale Imola	SC	*Sc/Sc*	*SC/SC*	*M/M*	[24]
121	Tondina di Costigliole	SC		*SC/SC*	*M/M*	
124	Big Red	SI		*SI/SI*	*M/M*	
134	Bebeco	SC	*Sc/S6*	*SC/SI*	*M/M*	[16]
136	Ouardi	SI	*S2/S7*	*SI/SI*	*M/M*	[16]
140	NJ A1	SC		*SI/SI*	*M/M*	[30]
145	Harleyne	SC	*S3/S20*	*SC/SI*	*M/M*	[33]
149	Sarritzu 1	SC		SC/SI	*m/M*	[31]
151	Petra	SC	*S1/S-*	*SI/SI*	*m/M*	[24]
152	Kyoto	SC	*Sc/S8*	*SC/SI*	*M/M*	[24]
153	Farmingdale	SC		*SC/SI*	*M/M*	[31]
157	Portici 1	SC	*S2/S17*	*SI/SI*	*m/M*	[24]
159	Bergecot	SC	*Sc/S2*	*SC/SI*	*M/M*	[14]
168	Lady Cot *	SC	*Sc/Sc*	*SC/SI*	*M/M*	[14]

**Table 2 genes-13-00548-t002:** Primers used in this study.

Primer	Sequence (5′ ≥ 3′)	Locus	Reference
*AprFBC8-F*	CATGGAAAAAGCTGACTTATGG	*S*	[13]
*AprFBC8-R*	GCCTCTAATGTCATCTACTCTTAG	*S*	[13]
*RFBc-F*	GAGGAGTGCTACAAACTAAGC	*S*	[17]
*RFBc-R*	ACCCCTATGATGTTCCAAAG	*S*	[17]
*SFBins-R*	TCAAGAACTTGGTTGGATTCG	*S*	[17]
*SFBc-F*	TCGACATCCTAGTAAGACTACCTGC	*S*	[11]
*SFBc-R*	ATTTCTTCACTGCCTGAATCG	*S*	[11]
*SFB-F*	TGGGTTCTGCAAGAAAAACGGTGG	*S*	This work
*SFB-OUT-R*	AATTCCTGTTTCAAGAACTTG	*S*	This work
*SFB-INS-F*	TTTTATGAGATTTTGGGGTTGGGC	*S*	This work
*SFB-INS-R*	GCCCAACCCCAAAATCTCATAAAA	*S*	This work
*SFBcj-F*	GTCCTTTTATTTAGAGATATTTAGTG	*S*	This work
*SFBcj-R*	ATAATCCGGAGGATAAATAAAAG	*S*	This work
*SFBj-F*	GGAGTAA/GCATACCACATTATTG-	*S*	This work
*Locus_M_F*	GGTGGTGGTCTAATGTGTTAAC	*M*	This work
*Locus_M_R*	TCCACTAGATCATGCTGCTT	*M*	This work

## Data Availability

All relevant data are available on reasonable request to the Corresponding Author.

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
