# Peer review of "Development of an HRMA-Based Marker Assisted Selection (MAS) Approach for Cost-Effective Genotyping of S and M Loci Controlling Self-Compatibility in Apricot (Prunus armeniaca L.)"

_genes, 2022, doi:10.3390/genes13030548_

Round 1
Reviewer 1 Report
I have the following (minor) comments:
Please include the name of the genus in the title.
The abstract should focus on the results and findings obtained in this study.
The manuscript needs to be carefully edited. Several sentences are misleading. For instance, line 30 “In the SI strategy, the self-pollen is rejected, and no pollen tube can grow to reach the ovule”. This is simply not true as pollen coming from other sources can grow. Self-pollen can also start to develop but then it stops. This is only example; there are many more sentences as this one in the text. Please revise the manuscript.
Please be more specific about which plants were used for the development and validation of PCR and HRM assays for both S and M loci, and used for assays, since selections and progenies might affect results.
Although the authors state they have developed the primers, I understand from Table 1 that some already existed from other studies. Please explain that better.
I don’t follow why amplicons were not sequenced.
“Discussion” appears twice: Results and Discussion; Discussion and Conclusions.
Author Response
Response to Reviewer 1 Comments
Point 1: Please include the name of the genus in the title.
Response 1: The name of the genus has been included
Point 2: The manuscript needs to be carefully edited. Several sentences are misleading. For instance, line 30 “In the SI strategy, the self-pollen is rejected, and no pollen tube can grow to reach the ovule”. This is simply not true as pollen coming from other sources can grow. Self-pollen can also start to develop but then it stops. This is only example; there are many more sentences as this one in the text. Please revise the manuscript.
Response 2: Thank you for your suggestions. The manuscript has been carefully edited to improve misleading sentences. Also the Introduction has been streamlined.
Point 3 : Please be more specific about which plants were used for the development and validation of PCR and HRM assays for both S and M loci, and used for assays, since selections and progenies might affect results.
Response 3: The 56 accessions used for the development and validation of PCR and HRM assay for both S and M loci, are listed in Table 1. The remaining plants were used for assay appllication and are listed in Supplemental File 2. A more precise specification was added to the manuscript.
Point 4: Although the authors state they have developed the primers, I understand from Table 1 that some already existed from other studies. Please explain that better.
Response 4: We firstly tested primers from Halàsz et al., 2010, Vilanova et al, 2006, and Romero et al, 2004. Unfortunately they were not suitable for our research purposes for different reasons (too large amplicons size or inability to amplify different alleles). Therefore, other primers were designed ex novo as indicated. Text was revised to further clarify this aspect, please see at line 120.
Point 5: I don’t follow why amplicons were not sequenced.
Response 5: Yes, it is a good practice the sequencing of amplicons. However, in our case, the specificity of obtained amplicons was supported by both electrophoresis gel and phenotypic co-segregation. Furthermore, we were not interested in characterizing additional polymorphism(s) within the target regions, as our scope was only the discrimination of transposon insertions at both loci.
Point 6: “Discussion” appears twice: Results and Discussion; Discussion and Conclusions.
Response 6: Fixed
Reviewer 2 Report
In this manuscript, authors demonstrated the feasibility of High Resolution Melting Analysis-based approach for rapid and cost-effective genotyping of self-compatibility at locus S and M to support apricot breeding. The work was done well, it is quite clear what results have been achieved.
Minor comments:
I believe that the text of 'Introduction' should shortened.
line 44: Apricot (P. armeniaca L.) is a diploid species... (not specie)
item 4 should be Conclusions (not Discussion and Conclusions).
Author Response
Response to Reviewer 2 Comments
Point 1: I believe that the text of 'Introduction' should shortened.
Response 1: You will find revisions in the next version of the manuscript
Point 2: line 44: Apricot (P. armeniaca L.) is a diploid species... (not specie)
Response 2: You will find the correction in the last version of the manuscript
Point 3: Item 4 should be Conclusions (not Discussion and Conclusions)
Response 3: Titles will be “Results and Discussion”, and “Conclusions”